# Developing interconnectedness is critical in retaining rural general practitioners: A qualitative thematic analysis of recently recruited general practitioners to South East New South Wales, Australia

**Sarath Burgis-Kasthala**[1,2], **Suzanne Bain-Donohue**[1]*, **Ellen Tailby**[1], **Kathryn Stonestreet**[1], **Malcolm Moore**[1]

**1** ANU Rural Clinical School, Acton, Australia Capital Territory, Australia, **2** University of St Andrews, North Haugh, St Andrews, Scotland

* suzanne.bain-donohue@anu.edu.au

**Data Availability Statement:** The data is available upon request. The data contains personal and

## Abstract

Australia, in common with many countries globally, has a shortage of doctors working rurally. Whilst strategies and current research focus on recruitment, attrition from rural practice is a significant determinant of such shortages. Understanding doctors' decisions to stay or leave, once recruited, may provide further insights on how to address this rural differential. This study comprises a qualitative study of 21 recently recruited nationally-trained doctors and international medical graduates to a rural area of New South Wales, Australia. Interviews focused on their experiences prior to and within rural practice, and how these influenced their future career intentions. We used reflexive thematic analysis with each interview coded by two researchers to build an explanatory framework. Our findings comprise five themes which applied differentially to nationally-trained doctors and international medical graduates: connectedness across professional, personal and geographic domains, how multi-faceted connectedness was, and dissonance between participants' expectations and experiences. Amongst nationally-trained doctors, connectedness stemmed from prior rural experiences which engendered expectations founded upon their ability to develop community-level relationships. Experiences were mixed; some described difficulties maintaining a boundary between their personal and professional lives, which encroached upon their ability to embed within the community. International medical graduates' expectations were cultivated by their pre-conceptions of Australian postgraduate training but they lamented a lack of professional opportunities once in practice. Moreover, they described a lack of professional relationships with local, nationally-trained, doctors that could help them embed into rural practice. This study highlighted that when connectedness occurs across professional, geographic and personal domains doctors are more likely to continue rural practice, whilst illustrating how the importance of each domain may differ amongst different cadres of doctor. Supporting such cadres develop supportive interrelationships may be a low hanging fruit to maximise retention.

identifiable material. To request access to the data please contact the Head of Rural Clinical School, Australian National University. Email:sally. hall@anu.edu.au.

**Funding:** The authors received no specific funding for this work.

**Competing interests:** The authors have declared that no competing interests exist.

## Introduction

A significant body of literature examines the shortages of doctors in rural areas. Whilst some literature separates the issues of recruitment and retention, much subsumes the two as both have overlapping factors (see Fig 1) which are commonly categorized into personal and professional domains.

At a personal level, some research has highlighted how certain personality traits, such as resilience, an acceptance of uncertainty, or appetite for novel experiences [1,26], are associated with choosing to work in a rural area. Moreover, individuals with a clearer interest or understanding of rural life, for example a specific desire to live rurally, or experiences of living rurally, are more likely to not only choose to work rurally but continue in rural practice [4,10–13,21,22]. This is intertwined with the needs and desires of their partners and families which places greater value on the characteristics of the rural community itself, such as its safety, employment opportunities and schooling, as well as the acceptance and integration of both the individual and their family within the local community [5,21]. Conversely, an interest in working rurally may be undermined by the potential disruption of existing personal and professional social networks.

At both a personal and professional level, several studies note that rural practice attracts doctors wishing a positive work-life balance while, at a professional level, some authors note that rural practice offers a greater opportunity to provide an extended range of services, greater continuity of care, and more autonomy, all of which can increase professional satisfaction [25,27]. These expectations may be fostered by doctors' prior experiences in both undergraduate and postgraduate attachments [1,4,13,14,19,20,28]. Conversely the ability to fulfill these opportunities may be contingent on doctors accessing professional support and several studies note that rural doctors have prominent levels of work-related stress compounded by professional isolation and a lack of professional networks [15,16,21–23,29]. This may especially apply to early-career doctors or students who require greater personal and professional support [1].

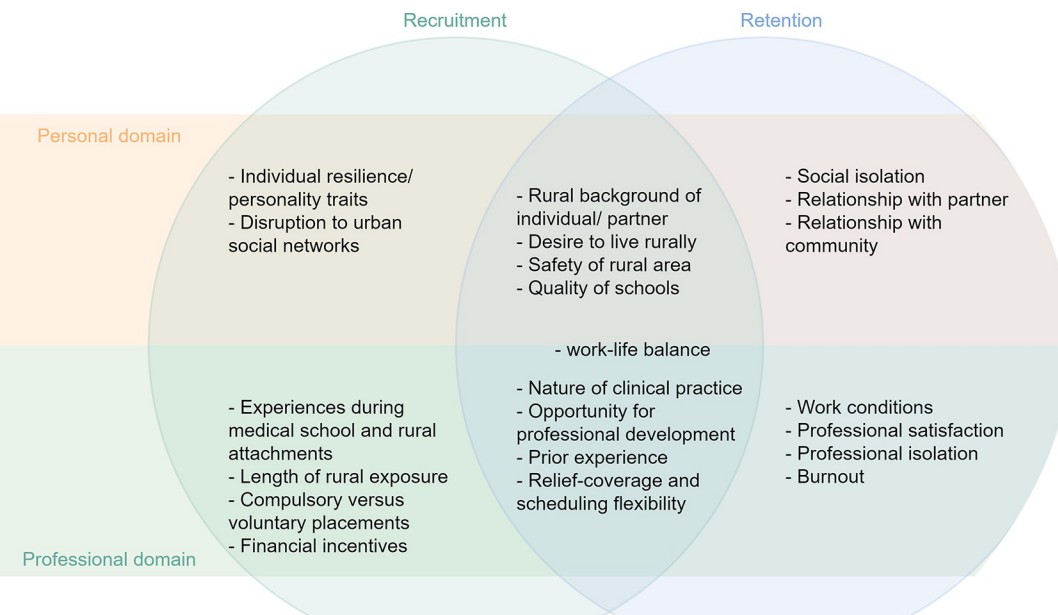

**Fig 1. Personal and professional factors associated with the recruitment and retention of doctors to rural practice [1–25].**

Most research examines recruitment with relatively little research examining retention or the continued success of interventions aimed at increasing recruitment [10,18]. The dearth of inquiry in retention is despite rural areas having higher health worker turnover, which both undermines the quality of healthcare and increases its costs [29,30]. A systematic review, while acknowledging the lack of research, also highlights the importance of a multifaceted approach addressing both pecuniary and non-pecuniary factors [18]. Moreover, empirical studies note that work stress, an adverse work-life balance in rural areas, and low job satisfaction may increase turnover [15], whilst some pecuniary incentives may help to address these [17]. These include: locum relief to facilitate doctors being able to take annual leave, retention payments for continued practice and additional payments which explicitly value added scope of practice [31]).

Accepting that the personality traits of individual doctors and previous affiliation either through living or medical education and training is a recognised driver of rural practice recruitment, the question remains as to what drivers promote rural retention, particularly after bonded or service requirement obligations have been met [1,4,10–13,21,22,26]. This article aims to identify factors in the retention of doctors over and above previously identified factors in the recruitment of doctors in rural locations. A deeper examination of factors that influence a rural medical career and long-term retention draws on the concept of connectedness and the interplay between different domains, which may illuminate elements, and strategies previously overlooked in addressing workforce shortage.

## Methods

### Ethics statement

Ethics approval was given by the Australian National University Human Research Ethics Committee (Protocol 2018/017). Formal written consent was obtained from all participants.

This study comprises a qualitative analysis of 21 semi-structured interviews utilising a reflexive thematic analysis [32]. This methodology was chosen, as it is a dynamic and iterative process that emphasises the importance of researcher engagement, reflection, and transparency. Reflexive thematic analysis involves identifying, analysing, and reporting patterns (themes) within the data.

### Setting

We invited doctors working in the 60 registered general practices in rural areas in South-East New South Wales, Australia excluding those who had worked in the region for more than 10 years. Doctors who has served in a rural area were deemed to have shown a commitment to rural practice and subsequently less of a risk to move to an urban setting. The geographical area covered both coastal and inland communities and aligned with the Australian National University Rural Clinical School footprint.

A key focus of health workforce policies in Australia has been addressing rural workforce shortages. This has led to the development of rural clinical schools which provide immersive rural educational experiences for undergraduate medical students in most universities in Australia [33]. More recently, this has been complemented by the introduction of integrated rural/regional training hubs which facilitate and provide opportunities for post-graduate training [34]. Specifically, informing the practice of international medical graduates (IMGs), are policies which obligate most to work in underserved areas of Australia, the majority of which are rural or remote [35,36].

**Table 1. Basic participant demographic information.**

| Origin | Male | Female | Total | Trained | Bonded/prescribed practice location |
|---|---|---|---|---|---|
| NTD | 4 | 6 | 10 | Australian Nation University (4)<br>University of Wollongong (3)<br>University of NSW (1)<br>University of Tasmania (1) | 4 |
| IMG | 6 | 5 | 11 | India (3)<br>Sri Lanka (2)<br>Pakistan (2)<br>Kenya (1)<br>Latvia (1)<br>United Kingdom (1)<br>USA (1) | 11 |

## Participants

The study utilised purposive sampling. Participants comprised 10 nationally-trained doctors (NTDs) and 11 IMGs working within the Southern NSW and Murrumbidgee Local Health Districts. Eight doctors practiced in inland communities whilst the remaining 13 practiced in communities in coastal areas. Participants had varying years of post-graduate experience. We initially contacted participants by sending an invitation together with a participant information sheet to their general practice (n = 60), with follow-up contact by email and telephone. Participant recruitment occurred from May to September 2018 (See Table 1).

## Data collection

We used a qualitative research methodology to explore the participant's experiences before and on entering rural practice, what the significant influences on their career were to date, and how these may influence their retention. After approved written consent was obtained we conducted semi-structured interviews by phone. Interviews ranged from 14 to 45 minutes in length and were audio-recorded. Confirmation of consent was obtained orally to ensure consent was willingly continued at the start of each interview, and transcribed verbatim by a third party transcription service bound by privacy protection requirements. The research team checked the transcriptions for accuracy. All data were de-identified to protect the confidentiality and privacy of all participants.

The semi structured interview covered issue related to demographics, journey to rural practice, influences in rural practice decisions, medical training background, supervision in current practice, positive and negative experiences and future aspirations (see S1 Annex).

## Data analysis

We utilised reflexive thematic analysis which combined aspects of inductive and deductive analysis [32]. The authors reviewed quantitative and qualitative research literature to develop an initial series of modifiable codes that comprised key concepts from Fig 1 (above). Data collection and analysis occurred iteratively. To ensure the capture of the richness and consistent management of data each interview was initially coded by hand, before two members of the research team re-coded it using NVivo11 [32,37]. Coding focused on the processes that encouraged participants to begin or continue in rural practice.

We used constant comparison to elucidate further codes, refining and/or subsume existing codes, revise our themes and inform our review of additional literature. We discussed these in a series of coding meetings aimed at developing a conceptual framework relevant to both NTDs and IMGs.

## Results

Our analysis provided a conceptual framework relevant to both NTDs and IMGs. The overriding concept which influenced doctors' intention to continue practicing rurally was how they developed connectedness. This comprised their relationships across personal, professional and geographic domains, and critically varied between nationally and overseas-trained doctors. It related to two further characteristics: the dissonance between their expectations and experiences, and if they developed connectedness holistically across all domains.

### Theme 1: Professional connectedness

Several NTDs described being attracted to rural practice by the positive relationships they had with their rural colleagues, who had often acted as their supervisors previously. They thus, as in this case, described a supportive practice environment that mitigated the negative aspects of the workload.

*we're doing so many different things and we do tend to work fulltime and after hours, but I guess what's kept me here is one, the supportive practice environment. (NTD10)*

Whilst NTDs described limited formal opportunities to develop further professional relationships, some described attending or organising informal activities which allowed them to develop a professional supportive community. For example, this participant described arranging informal activities between other registrars.

*I try to keep that in mind and organise just informal things between the registrars and keep tabs on where the new registrars are in the area and try and link them up with the other registrars, but I think that's not really done that well otherwise by their training organisations. (NTD07)*

Several IMGs described how they had been recruited into their current rural practice by an existing IMG who had supported their arrival and introduced them to their practice. As in this case, this support was often fostered by their common training needs. Thus, rather than being necessarily attracted to the rural location, IMGs highlighted their professional reasons for working rurally. They, as in this case, often chose to work in a specific location based on the advice from or relationship with another IMG who could help orientate them with Australian medical practice.

*So I am the one who brought him here. I ask him to come and join with me because I told him I will be alone so you come here, we'll study together. We will do a group study in the evening. We will do a research. We will work together and it would be good for me to be able work together, prepare for the fellowship.(IMG18)*

In contrast, several IMGs stated that they did not feel welcomed by existing NTDs which contributed to a sense of isolation and professional detachment. This tended to occur in specific locations in which practices within towns were divided between those comprising NTDs and those comprising IMGs. Even when they felt generally welcomed, IMGs described having few opportunities to meet local doctors within the area and form professional relationships. This is described clearly by this participant.

*Other doctors I haven't meet, met a lot of people here. Only I attend one meeting in the [town] hospital when I came here first time. Then we went week ago for second meeting but*

*unfortunately they cancelled the meeting for some reason. So I don't know other people or doctors here a lot. I don't meet with them. (IMG18)*

Instead, IMGs described how their professional networks were spatially widespread. They often used these to study and were centred around passing the requisite exams to allow continued practice in Australia.

*No no it's through social media, the Skype, the Facebook and the, there are some we meet in the, on conferences so we had the common interest and we you know, some are going through the same, they are going to give the same exams. So just exchange the numbers and we will just start studying together and form a small study group which help each of us. So and that way we are able to you know exchange the knowledge and whatever information we have. (IMG11)*

Whilst the majority of NTDs described having professional relationships with peers or other health professionals, very few discussed their professional relationship with their patients and how this impinged upon their intention to continue rural practice. In the few circumstances they did, the focus was managing the integrity of their personal lives in a rural environment. This is highlighted in the following quotation.

*I went from a town of 1,300 people to a town of 10,000 people, and the patients were immediately more demanding and less giving, and they thought it was okay to call you on your home phone just because they thought that was okay to do, the people [smaller town] would not, they had a much greater respect for leaving you at home to your family without interrupting you. (NTD05)*

In contrast, many IMGs descriptions of working with local communities were centred around a sense of altruism and meeting the unmet, often significant, needs of their patients. IMGs ubiquitously described the positive reaction they had from local communities which enabled them to form a professional relationship with their communities. They often contrasted this with the limited acceptance they had received from their Australian trained colleagues, and was thus especially significant and had greater meaning than for NTDs who expected to be welcomed professionally and by their communities.

*The people are, so I mean they appreciate, you know, people coming to, doctors coming to the small town and you can feel that. You know my, they won't gripe about waiting for 45 minutes in the waiting room and not, you know they're still happy. We're welcome. (IMG15)*

### Theme 2: Personal connectedness

Nationally trained doctors described expecting to find a workplace that supported their needs to maintain their work-life balance, and a community in which they could be part of, and how these expectations were largely met by their experiences.

*It's a very active town, and it's a reasonably small town, so it's about 1,500 people, and so living here as a GP trainee you get to know people really quickly, and the lifestyle is amazing, so I would surf, ride my bike, be on the beach multiple times a week, and the attitude towards young adults moving here is really positive, so the town just loves seeing young adults move here, put their roots down, so they blend into the community pretty quickly. (NTD07).*

Many consequently stressed how they had developed personal relationships within their communities. Critically these allowed them to embed into their rural community and increased the likelihood that they would stay as described in the following quote.

*I play in a folk band in town and I also have been playing soccer with one of the local women's teams and I guess through that they're my non-work related social networks and I guess they have helped to provide a balance to the medical side of things and I think that those external connections are also staying factors that would make me more likely to stay than go.(NTD10)*

Several NTDs described the importance of their partners also finding work or settling into rural life. Some intimated that this was due to their or their partner's personality traits, as well as holding rural interests, and being flexible in changing career or being a primary carer. In the following quote a doctor describes how her partner had to change career and take on more childcare responsibilities following their move from a city location.

*No, he's more than happy working in admin. I think he's fairly laid back anyway so it works quite well for him and we can sort of juggle the girls between us, which is good. But yeah, I know a few of my friends have come down and not been able to find work for their partners. (NTD01)*

IMGs often described the importance of their immediate family and how this comprised their main personal social network. Conversely, IMGs described developing few local relationships asides from their family, and the professional relationships they had with their patients. In some cases, participants described intending to move to a larger urban location in which they could develop more community-orientated relationships.

*They're very good, they're very helpful. I like the patient population a lot. Flexibility is huge. You know they accommodate to you know most things in terms of my limitations because I don't have any social support here. Which is a, which is another big challenge. (IMG14)*

## Theme 3: Geographical connectedness

NTDs described clearly how they were drawn to rural Australia, and often the specific area they worked in.

*But, you know, I suppose to go out and be a doctor in a small country town I think you need not only a commitment to medicine but a bit of a sense of adventure and certainly a like of outdoors-type activities. (NTD05)*

Several NTDs highlighted how rural practice was compatible with a life beyond medicine, and a focus on their own well-being. Often this was related to aspects of living rurally, such as outdoor pursuits. For example, this participant described the pull of his rural coastal practice.

*So then I did medicine with no aspirations to be a GP, but I enjoy my outdoors. I'm a surfer, most of all, and so as I went through medicine and then, particularly when I did my hospital years, I yearned for this idea of living on the coast and surfing. (NTD08)*

The search for a compatible location also helped NTDs differentiate between different rural towns, whilst balancing geographical remoteness and access, for example to the coast, and

their connectedness to other towns and their existing social networks. Together, as described in this quote, this influenced their choice of practice.

*Just the geography, so I really enjoyed working in [small coastal town]. So basically I was working in [coastal town] originally and then [small coastal town] had a severe shortage of GPs, so then I was helping them out for six months, and it's about a 35 minute drive, but it's quite windy, there's lots of wildlife, it's beautiful but a difficult drive, and then I realised that I really liked that smaller practice, two GPs in a practice, a single practice town feel, and the medicine there, so I would've stayed there if I lived closer to [small coastal town], basically, but because I didn't like the drive I'm about to stop working there and I'm now starting working in [bigger coastal town] instead because that will be very a similar size practice to the [small coastal town] practice, but it's just closer to home. (NTD07)*

In contrast to NTDs, the majority of whom had chosen to have rural experiences and work rurally, most IMGs had chosen to work in Australia and, due to the policy of overseas doctors having to work in an underserved area, were then required to work rurally. This difference was apparent in their discussions about the nature of their work, as in this case.

*Oh no it's personality. Yeah nothing to do with the work. The work's really interesting, there's a lot of sick people that live rurally. So it's very interesting, it's just not you know—personally I'm a city person.(IMG14)*

Their responses, thus, described their broader appreciation of Australia as opposed to specific rural areas. They often contrasted this with their prior experiences of working internationally as in this case of an IMG who had previously worked in Canada.

*We are happy here in Australia. People are very good. There's no discrimination, there's no, we haven't been victimised or harassment or discrimination or anything like that. And people are quite supportive. In Queensland people are wonderful. I found them they never been, I never, I stay there for two year and plus two months then never had any even minor issue there so. And weather is pretty good. No snow in Australia. I don't need to shovel in the morning my driveway. So we want to stay here, we don't know. We love Australia, we love this country. (IMG18)*

This was consistent with how they described the spatial breadth of their relationships. Whilst NTDs described developing personal, professional and geographical connectedness at a local level, IMGs often described having a wide set of professional and personal relationships spanning different regions and countries. Whilst both IMGs and NTDs signified the importance of local amenities in their choice of work location, IMGs specifically highlighted the absence of amenities, such as private education for their children, not available rurally.

*it's really hard to bring doctors in this place because most of them either they want to settle down close to places where all the amenities are available. Like they're looking for education of the kids and they're looking for like town life. (IMG21)*

### Theme 4: Dissonance between expectations and experiences

Many NTDs described prior rural experiences, either due to having a rural background, or from undergraduate or postgraduate rural placements. These engendered a series of

expectations, at personal and professional levels, primarily relating to their envisaged scope of practice and work-life balance. Most NTDs described their professional expectations being met, especially within slightly larger teaching practices, as described in this case.

> *When I was here as a student I got to see a lot and do a lot and I thought this would be a good place to practice eventually because and I'm still learning currently. So they have good supervision, good colleagues, good amount of support from the hospital and from visiting specialists and things like that. But still with the ability to really put the responsibility on me and allow me to kind of do what I'm comfortable with and kind of make my own decisions. So it gives you that good balance between independence and support. (NTD02)*

In contrast some NTDs noted that, despite their prior rural experiences, once they began to settle into rural life, their experiences contrasted with some of their idealistic expectations. Specifically, some felt disconcerted as the boundary between their personal and professional lives was transgressed. For example, in the following quotation, a doctor describes the claustrophobia her partner felt which constrained him from simple activities like grocery shopping.

> *So I think it's probably that privacy that most people would struggle with. My partner found that a little bit unusual to start off with but I guess I was just used to it being from a small town. Yeah, so that's probably the one and some people find that's fine, but I think a lot of doctors struggle a little bit with that. I know [partner] doesn't go out before 10:00 to do his shopping because he tries to avoid running into people, more for their embarrassment than his. (NTD01)*

IMGs conversely, highlighted entering rural practice as a rite-of-passage to allow them to obtain vocational qualifications. Many had then envisaged moving to a more urban location. They thus had specific expectations of rural practice, which were centred around the professional support and opportunities they could obtain. They often described that such support, which limited their opportunities to obtain work, was limited.

> *You expect that you'll go there, and you'll get a job very quickly, it's a nightmare to find a job as a GP here. So, I applied almost 1,000s of different practices when I came here, and no one basically tries to understand the problems that we're facing. (IMG21)*

### Theme 5: Developing multi-faceted connectedness

Participants described how their relationships across personal and professional levels were key to how they evaluated their expectations and experiences, and obtained and utilised opportunities for professional and personal growth. Significantly, for both cadres, it was connectedness across domains that were more likely to lead to their retention.

Some participants noted how personal isolation may compound professional isolation, and thus highlighted the importance of ensuring rural areas had networking processes and meetings to provide support.

> *So and you know during, the nature of general practice is during the day you do kind of work by yourself and you know during your lunch break you might chat to one of your colleagues or something. Or you might meet them for a meeting but in general the work is in and of itself you're by yourself. And so if you compound that with then living in a rural area and not having that many friends or something like that, it can become a bit difficult. So and it's not, so*

*the isolation part isn't for everyone. Especially if you've recently moved to a rural town. And I guess the way to tackle that would be to just continue having a lot of those meetings and networking events and support systems. (NTD02)*

Some participants described how negative aspects, such as the workload, of their role could be mitigated by their relationships at a community-level. For example, this doctor describes continuing to work in challenging conditions as the community banded around to try and mitigate the overload. Thus connectedness at a professional, community-level, mitigated some of her isolation, albeit only temporarily, before she left for another rural post.

*I very much was made to feel that I was an important part of the community and they were keen to help me in any way they could. And, you know, they did, and they tried not to come in after hours and, you know, people did make an effort, and that made me less likely to leave, of course. (NTD05)*

Finally, some described the interrelation and importance of the characteristics of the town as helping to engender broader personal connectedness for both participants and their partners and families. For example, this NTD described moving to a town which had the right size and diversity to enable both he and his partner to develop adequate relationships within the community. Critically, this was also dependent on their own expectations and personal needs for such relationships.

*Oh, this town is very good from a social point of view, strangely enough. Everyone's very friendly. I think it's just a good size with enough young people that you're a new young person. That does seem right, yeah. Immediately, we made a handful of friends and I only need about two or three friends. In fact, three's probably getting to overload. I'm happy, yeah. (NTD08)*

This also applied to developing adequate professional connectedness as described by another NTD.

*Also I wanted to go rural but didn't want to go remote. I didn't want to be in a one or two doctor town where I was relied on, basically. So somewhere like [town] is really good because there's heaps of doctors here. (NTD06)*

## Discussion

Drawing on discourses of social capital and the theoretical framework of social constructionism this study proposes that the overarching determinant of retention is doctors' connectedness across personal, professional and geographical domains and subsequent strength of interconnectedness between domains whilst in rural practice, and how this may be inhibited by a dissonance between their expectations of rural practice and their experiences since recruitment. This is supported by other research, for example a systematic review of nurses' decisions to work rurally highlighted the influence of place, professional and personal factors [38]. Moreover, Schoo et al's (2016) conceptual framework illustrates how social capital at individual, community and organizational levels–comprising both personal and professional relationships–is critical in ensuring cohesive health services that can both recruit and retain their health workforce [39]. Social constructionism identifies the perceived reality of an individual is the product of dynamic interactions with social conventions and structures whereby their understanding and positioning of self are constructed [40]. Whilst the definition of

connectedness is variable depending on the research area and theoretic model utilized for analysis this study defined connectedness as the feeling of a deeper connection or bonding with an individual's sense of self, their profession or workplace, and their lived environment. Moreover, in our study, connectedness occurred at different sociorelational and geographical levels, for example, with those from the same rural location, in proximal locations, or with those from distant locations. Key to understanding this interaction were our comparative analyses of the perspectives of nationally-trained doctors and international medical graduates.

NTDs ubiquitously had rural experiences prior to working in rural SE NSW, either from childhood, through undergraduate and/or postgraduate placements, or their previous work as a general practitioner. These engendered a series of expectations largely founded on their ability to develop community-level personal relationships within their rural location. Whilst many described being able to do so, some had described how specific rural locations were more likely to meet these needs. Moreover, some described difficulties maintaining a boundary between their personal and professional lives, and how this encroached upon their ability to be part of the community. This is supported by Bentley et al's (2019) cross-sectional study which demonstrated how graduates of Australian rural clinical schools had cultivated a series of expectations that encompassed their professional practice, as well a personal and professional interconnectedness [41]. Significantly, whilst expectations regarding professional practice were largely surpassed, a substantial minority of graduates stated that their expectations of personal and professional support were not met with this gap between expectations and experience associated with a reduced intent to practice.

Critically, NTDs in our study described relationships which were spatially confined to their local area. Often these were intertwined with geographical aspects of rurality, such as enjoying outdoor activities. Thus, when they developed further relationships, they became more embedded rurally. Conversely, if they did not develop such relationships, their absence of spatially distant relationships meant they were more likely to depart.

IMGs' expectations, in contrast, were cultivated by their pre-conceptions of Australian postgraduate training, often informed by their IMG peers with whom they had developed spatially broad personal and professional social networks. Often these were crucial to their recruitment as they had often travelled from outside Australia to work in their rural location. Their recruitment was accompanied by expectations of being able to enter training programmes and/or gain professional qualifications that could increase their professional mobility to work in other, more urban, locations within Australia. Some IMGs described local relationships with other IMGs in their practice, who had often been integral to their recruitment; these relationships often provided a degree of professional support. Aside from these relationships, their most distinctive local relationships tended to be their professional relationships with their communities and patients. In contrast, the majority of IMGs interviewed described developing limited relationships with local doctors and none described forming personal relationships with other members of the community. They did, however, describe feeling accepted by their communities and being made to feel welcome. The importance of this is supported by other research examining IMGs' experiences in rural Australia.

Interestingly, NTDs' and IMGs' descriptions of their personal community relationships were different. This contrast may be explained by their differing expectations. NTDs expected to form personal relationships within the community thus were sensitive to overlap between professional and personal community relationships. In contrast, IMGs had limited expectations from local communities; thus the positive professional relationships were welcome with less scope for these transgressing their personal lives. This is supported by other research from Australia which highlighted that IMGs' expectations of their communities were limited to a sense of acceptance rather than stronger personal and individual relationships [42,43] These

differences in expectations may be related to the different cadres' of doctors intentions upon beginning rural practice: NTDs often intended to stay whereas most IMGs envisaged staying rurally for a limited period of time only. They might not have seen developing personal connectedness as beneficial which may dually have limited their expectations.

Unlike the majority of NTDs, several IMGs described strong professional supportive relationships and personal relationships with individuals across Australia or internationally, and many used virtual means of communicating. Moreover, many had moved as a family unit. Thus, whilst a lack of personal relationships with those within their community limited how some participants embedded rurally, they were able to compensate for this lack of local connectedness by communicating across longstanding spatially distant, often virtual, social networks.

Essentially, if a doctors' connectedness occurs across all personal, professional and geographic domains with the majority derived from their rural location, then the doctor is likely to feel connected and embedded within their rural location increasing retention. In contrast, if some aspects of their connectedness are undermined by rural practice, as was the case when NTDs described their personal boundaries being transgressed, this may limit retention. Conversely, the presence of strong virtual social networks, developed over years through travel across multiple locations, amongst IMGs potentially allowed them to compensate for this lack of connectedness. However, this link between connectedness and retention is more friable; whilst some spatially-distant connections may mitigate a lack of connectedness in one or more domains, if the minority of relationships are within the rural location there may be no compelling reasons for doctors to stay. Critically, almost all IMGs described a lack of professional relationships with local, nationally-trained, doctors. These may be categorically different from their relationships with IMGs, by providing greater opportunities to develop professional networks and opportunities which would help them embed into rural practice. Thus, increasing professional support may be low hanging fruit to increase retention of IMGs. Supporting this, while mandated IMGs have high rates of job dissatisfaction [36], an example from an Australian region in which, in contrast to most other parts of Australia, IMGs were able to gain accreditation demonstrated IMGs with higher rates of job satisfaction and perceived professional support than their nationally-trained colleagues [44].

By corollary, another retention strategy might be to develop virtual networks in NTDs. Interestingly, some training programmes have sought to do this–principally to improve their efficiency whilst mitigating the difficulties of travelling large distances between remote practices [19]. Barnett et al (2012) identify that self-selection in virtual communities is more likely to yield greater participation and therefore networking benefits [45]. Even passive members of virtual networks derive benefit whilst those within a network that are more knowledgeable and share their knowledge, gain social capital [45]. However, there are two flaws with this approach. Firstly, none of the NTDs in our study indicated a wish to have broader virtual networks; if anything, they highlighted a greater need for local opportunities to meet their peers and colleagues. Secondly, virtual initiatives may replace local opportunities and developments. IMGs, by virtue of their pathway to Australian rural general practice, described spatially distant social networks as a consequence and evolution of their migratory lives. An analysis of IMGs in Australia highlighted how such approaches may be a coping mechanism, as they have to prioritise their mandated work requirements46). This however, may prevent them from embedding into rural life as the uncertainty of their future practice prevents them from settling into rural practice. Thus, on completion of their mandated work, lifestyle and family priorities may, in the majority of cases, lead to their migration to urban practice. This was partly seen in our study; as IMGs spent greater time rurally some lamented a lack of personal relationships,

either with peers or other community-members, which discouraged from continuing to practice rurally.

Recruitment strategies should maximise opportunities for practitioners to embed rurally. This may be by selecting doctors with personality attributes, such as having a rural identity and having high cooperativeness, associated with rural practice, and by ensuring training programmes are geographically circumscribed to facilitate practitioners, and their families, building relationships locally [46,47]. This is also relevant to IMGs who, rather than being seen as a short-term stopgap, could be offered longer-term opportunities, inclusive of professional development, to practice rurally [48].

This study has several limitations. It includes the perspectives of a limited number of doctors from one region in South-East New South Wales only. More specifically, whilst all doctors work in rural locations, and some alluded to more remote work, our analysis did not explicate the importance of how remote their work was and how this impinged upon their connectedness and decision to continue in rural practice. Moreover, whilst IMGs are an important facet of the health workforce globally, in Australia most IMGs are constrained by specific work conditions which limit where they may practice. As a result, some descriptive findings of this study are limited in their wider applicability. We contend, however, that more conceptually, the importance of connectedness and how this resonates across the different domains of the doctors' lives, transcends these descriptive findings and is a useful tool in understanding how we can better develop policies that maximise both recruitment and retention.

## Conclusion

This study presents a qualitative analysis of how recently recruited rural general practitioners developed connectedness and how it influenced their thoughts on continuing rural practice. It highlights the importance of doctors developing connectedness across geographic, personal and professional domains. We also highlight differences between nationally-trained doctors and international medical graduates whilst supporting greater opportunities for nationally-trained doctors to embed rurally and form synergistic professional networks with international medical graduates.

## Supporting information

**S1 Annex. Semi-structured interview questions.**
(DOCX)

## Author Contributions

**Conceptualization:** Sarath Burgis-Kasthala, Ellen Tailby, Kathryn Stonestreet, Malcolm Moore.

**Data curation:** Sarath Burgis-Kasthala, Suzanne Bain-Donohue, Ellen Tailby, Kathryn Stonestreet.

**Formal analysis:** Sarath Burgis-Kasthala, Suzanne Bain-Donohue, Ellen Tailby.

**Investigation:** Ellen Tailby.

**Methodology:** Sarath Burgis-Kasthala, Suzanne Bain-Donohue, Ellen Tailby, Kathryn Stonestreet, Malcolm Moore.

**Project administration:** Sarath Burgis-Kasthala, Suzanne Bain-Donohue.

**Supervision:** Malcolm Moore.

**Validation:** Sarath Burgis-Kasthala, Suzanne Bain-Donohue, Ellen Tailby.

**Visualization:** Sarath Burgis-Kasthala, Suzanne Bain-Donohue.

**Writing – original draft:** Sarath Burgis-Kasthala, Suzanne Bain-Donohue, Malcolm Moore.

**Writing – review & editing:** Sarath Burgis-Kasthala, Suzanne Bain-Donohue, Ellen Tailby, Kathryn Stonestreet, Malcolm Moore.

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
