## [Decision Letter · Decision Letter 0]

16 Oct 2023

PGPH-D-23-01460

Developing interconnectedness is critical in retaining rural general practitioners: a qualitative thematic analysis of recently recruited general practitioners to South East New South Wales, Australia.

Dr Suzanne Bain-Donohue

Thank you for submitting your manuscript to PLOS Global Public Health. After careful consideration, we feel that it has merit but does not fully meet PLOS Global Public Health’s publication criteria as it currently stands. Therefore, we invite you to submit a revised version of the manuscript that addresses the points raised during the review process.

We look forward to receiving your revised manuscript.

Kind regards,

Henry Zakumumpa, PhD

Academic Editor

Journal Requirements:

1. In the ethics statement in the Methods, you have specified that verbal consent was obtained. Please provide additional details regarding how this consent was documented and witnessed, and state whether this was approved by the IRB.

Additional Editor Comments (if provided):

We are pleased to share comments from our reviewers. Please pay attention to each of the comments raised by the reviewers. I wish to point you to particularly comments regarding the framing of the paper and the methodology. Observations that the authors 'used thematic analysis for quantitative data' are especially problematic.

Reviewers' comments:

Reviewer's Responses to Questions

**Comments to the Author**

1. Does this manuscript meet PLOS Global Public Health’s publication criteria? Is the manuscript technically sound, and do the data support the conclusions? The manuscript must describe methodologically and ethically rigorous research with conclusions that are appropriately drawn based on the data presented.

Reviewer #1: Yes

Reviewer #2: No

2. Has the statistical analysis been performed appropriately and rigorously?

Reviewer #1: N/A

Reviewer #2: I don't know

3. Have the authors made all data underlying the findings in their manuscript fully available (please refer to the Data Availability Statement at the start of the manuscript PDF file)?

Reviewer #1: Yes

Reviewer #2: Yes

4. Is the manuscript presented in an intelligible fashion and written in standard English?

Reviewer #1: Yes

Reviewer #2: Yes

5. Review Comments to the Author

Reviewer #1: Thank you for the invite to review the article.

I very much appreciate the authors for an interesting research and paper.

Please find a few minor and major suggestions to help improve the quality of the paper.

Minor

Line 78, as part of the methods, please describe the overarching research design.

Line 82, 'we invited ALL doctors', specify how many.

Line 82, 'excluding these', specify how many.

Table 1, Perhaps just mention the broad categories in the text, and move this table to annex.

Line 325-335, Please focus on the dissonance aspect and move the content to any of the aforementioned appropriate themes. For now it comes across as repetition/redundant.

Major

Line 137, 'meetings aimed at developing a conceptual framework', I think it is worth sharing a visual of the framework as opposed to the above table with the interview instrument.

The study seems to draw on discourses of social capital, judging from the focus in framing the findings around interconnectedness across three domains. Perhaps presenting explicitly the underlying framework along with sufficient description can make the paper richer.

The use of theory is also warranted (e.g. social capital or personality related) to shed further light on the findings about the role of the three interconnectedness domains.

I find the notion of 'doctors interconnectedness' a bit vague perhaps due to lack of a theoretical grounding. Picking on your analysis about dissonance in expectation and actual experience, I wonder if interconnectedness can be rephrased as 'alignment in expectation and experience of doctors across the three domains'?

The findings suggest that retention is contingent on personal, professional, OR geographic interconnectedness, with some differentiated patterns between the two groups of doctors. Most importantly decision to stay or leave seem to be mediated by personality or personal circumstances.

Related studies seem to underscore the significance of personality. See two related sources for your reference or integration:

Rural temperament and character: A new perspective on retention of rural doctors, by Eley, Young, Shrapnel

Is personality the missing link in understanding recruitment and retention of rural general practitioners? by Jones and Humphreys

Please also shed light, if data allows, of any differentiated pattern based on demographic factors (age, gender, family/marital status- which may be closely related to personality).

Best wishes

Reviewer #2: The authors have written a paper on a topic that is of interest to them which comes across in the article which needs to be reframed to provide an academic contribution. For example, the authors say “Axiologically, the team shared an explicit interest in understanding to increase the retention of doctors working in rural areas and this was articulated in the written participant information sheets we provided.” The article would be improved if the authors reframed the paper for the reader, e.g. The aim of the research was to … and then provided academic reasoning for the need for the research they have conducted. To this point, the gap in the literature to which this study contributes needs to be presented clearly prior to the methods section.

The methods also need to be revised. The participant section needs more detailed information, e.g. how did you identify the potential sample for your study? The participant information is hard to follow written in this way, please consider putting it into a table. Then, the quotes make more sense to the reader, e.g. is the quote from a rural doctor who has been there for 1 year or 7 years? The authors had an interest in retention yet they excluded those that been there more than 10 years – why? The exclusion criteria needs to be explained. Also the section about the researchers feels unnecessary. The authors say that they sent a participant information sheet but then say they had verbal consent to participate in the study. Was verbal consent approved by the HREC, this seems inconsistent with usual practice in Australian universities.

In terms of data collection, a reflexive thematic analysis was conducted on quantitative and qualitative data. It is unusual to use a qualitative data analysis technique to analyze quantitative data. The data was initially coded by hand and then coded with NVIVO. It is unclear why this was done in this way. The methodology needs further explanation with references to support the chosen data analysis techniques chosen for their data.

Once the authors have provided an aim and rationale for the study, and revised the methods they will be in a better place to present the findings aligned to the research aim.

Of note, there are a lot of comparisons between NTDs and IMDs, this appears important to the findings but not positioned in the frontend of the article as being purposely investigated.

If there authors were to revise the entire article it may be suitable for publication.

6. PLOS authors have the option to publish the peer review history of their article (what does this mean?). If published, this will include your full peer review and any attached files.

**Do you want your identity to be public for this peer review?** For information about this choice, including consent withdrawal, please see our Privacy Policy.

Reviewer #1: **Yes: **Woldekidan Amde

Reviewer #2: No

---

## [Decision Letter · Decision Letter 1]

15 Feb 2024

Developing interconnectedness is critical in retaining rural general practitioners: a qualitative thematic analysis of recently recruited general practitioners to South East New South Wales, Australia.

PGPH-D-23-01460R1

Dear Suzanne Bain-Donohue,

We are pleased to inform you that your manuscript 'Developing interconnectedness is critical in retaining rural general practitioners: a qualitative thematic analysis of recently recruited general practitioners to South East New South Wales, Australia.' has been provisionally accepted for publication in PLOS Global Public Health.

Best regards,

Henry Zakumumpa, PhD

Academic Editor

Thank your your efforts in responding to our reviewers' comments.

Reviewer Comments (if any, and for reference):

Reviewer's Responses to Questions

**Comments to the Author**

1. If the authors have adequately addressed your comments raised in a previous round of review and you feel that this manuscript is now acceptable for publication, you may indicate that here to bypass the “Comments to the Author” section, enter your conflict of interest statement in the “Confidential to Editor” section, and submit your "Accept" recommendation.

Reviewer #2: All comments have been addressed

2. Does this manuscript meet PLOS Global Public Health’s publication criteria? Is the manuscript technically sound, and do the data support the conclusions? The manuscript must describe methodologically and ethically rigorous research with conclusions that are appropriately drawn based on the data presented.

Reviewer #2: Yes

3. Has the statistical analysis been performed appropriately and rigorously?

Reviewer #2: N/A

4. Have the authors made all data underlying the findings in their manuscript fully available (please refer to the Data Availability Statement at the start of the manuscript PDF file)?

Reviewer #2: (No Response)

5. Is the manuscript presented in an intelligible fashion and written in standard English?

Reviewer #2: Yes

6. Review Comments to the Author

Reviewer #2: The authors have substantially revised their manuscript and it is reading a lot better; however, it could benefit from further attention. Specific items to review are noted below. In addition, I would suggest that the authors consider their use of quotes are how well they exemplify the point that they are trying to make in their article. Also, some of the unedited quotes are difficult to follow in their current format. In addition, while it is not essential for publication, the article would be improved with a clear description of the themes and the key terms, e.g. it would help the reader to know how the authors conceptualize the term ‘interconnectedness’ earlier in the article.

Page 4, Line 74-76:

Please consider revising this sentence, it is either missing punctuation or words, “These include: locum relief to facilitate doctors being able to take annual leave, retention payments for continued practice and additional payments which explicitly value added scope of practice(31).”

Page 4, Line 94:

There is a typo, “Doctors who has served in a rural area were deemed…”

Page 6, Line 124:

There is a typo, “The semi structured interview covered issue related to demographics …”

The use of third person for academic writing is preferred over first person, e.g. page 6, “We utilised”, “We used”

Page 8, line 179:

There are large quotes that do not flow well and are difficult to read, for example on page 8, line 179 the authors say “This is described clearly by this participant” and then provide a quote that is difficult to understand.

“Other doctors I haven’t meet, met a lot of people here. Only I attend one meeting in the [town] hospital when I came here first time. Then we went week ago for second meeting but unfortunately they cancelled the meeting for some reason. So I don’t know other people or doctors here a lot. I don’t meet with them. (IMG18)”

I understand that the authors are quoting participants directly but please consider how to better present the quote or consider an alternative quote that better supports your point.

Page 13, line 311:

Please consider the use of claustrophobia in this sentence because the quote describes a lack of privacy or boundaries, rather than a condition such as claustrophobia, “a doctor describes the claustrophobia her partner felt…”

Page 15, line 392:

Please check reference is correct “interconnectedness(3941).”

Page 16, line 421:

Please check reference is correct “relationships(4042)”

Page 18, line 452:

Please check reference is correctly punctuated “benefits45).”

Page 18, line 453:

Please check reference is correctly punctuated “gain social capital45)”

Page 18, line 459

Please check reference is correctly punctuated “work requirements46)”

In the reference section, please check the formatting for Reference 37 and 38

In the reference section, please check the formatting for Reference 42

7. PLOS authors have the option to publish the peer review history of their article (what does this mean?). If published, this will include your full peer review and any attached files.

**Do you want your identity to be public for this peer review?** For information about this choice, including consent withdrawal, please see our Privacy Policy.

Reviewer #2: No
